# Precision of an Inertial System to Evaluate the Finger Tapping Test in Women with Fibromyalgia

**DOI:** 10.3390/sports13110373

**Published:** 2025-10-31

**Authors:** Nancy Brígida, David Catela, Cristiana Mercê, Marco Branco

**Affiliations:** 1Sport Sciences School of Rio Maior (ESDRM), Santarém Polytechnic University, Av. Dr. Mário Soares, 110, 2040-413 Rio Maior, Portugal; catela@esdrm.ipsantarem.pt (D.C.); cristianamerce@esdrm.ipsantarem.pt (C.M.); marcobranco@esdrm.ipsantarem.pt (M.B.); 2Sport Physical Activity and Health Research & Innovation Center (SPRINT), Santarém Polytechnic University, Complex Andaluz, Apart 279, 2001-904 Santarém, Portugal; 3Physical Activity and Health—Life Quality Research Center (CIEQV), Santarém Polytechnic University, Complex Andaluz, Apart 279, 2001-904 Santarém, Portugal; 4Quality Education—Life Quality Research Center (CIEQV), Santarém Polytechnic University, Complex Andaluz, Apart 279, 2001-904 Santarém, Portugal; 5Interdisciplinary Center for the Study of Human Performance (CIPER), Faculty of Human Kinetics, University of Lisbon, Cruz Quebrada-Dafundo, 1499-002 Lisboa, Portugal

**Keywords:** inertial measurement units, finger tapping test, Bland–Altman, motor control, fibromyalgia

## Abstract

Background: This study investigates the precision of an inertial measurement unit (IMU) in evaluating the Finger Tapping Test (FTT) to differentiate motor control competencies in women with fibromyalgia, a clinical population characterized by motor impairments. Methods: The sample consisted of 240 FTT trials collected from 20 women, half of whom were diagnosed with fibromyalgia (F = 46.4 ± 12.714; C = 45.9 ± 12.950). Procedures consisted of participants completing FTT while data were collected from a high-speed camera and an IMU for linear acceleration and angular velocity, respectively. Analyses employed the Bland–Altman technique with both parametric and bootstrap-derived limits of agreement and intraclass correlation coefficients to assess levels of agreement between traditional and IMU-derived methods. Results: The results showed a strong agreement at subject×hand aggregation for the number of taps (RPC = 4.3 and ICC = 0.94) and for the inter-tap interval (RPC = 0.02 and ICC = 0.89), indicating minimal differences between measurements and demonstrating the potential for highly sensitive motor function assessment using an IMU. Conclusions: These findings suggest that IMU technology can effectively detect subtle aspects of motor control, supporting its use in exercise, rehabilitation, and clinical physiotherapy settings, including functional training, adapted rehabilitation exercises, and home-based monitoring for fibromyalgia. This approach offers detailed insights into subtle motor impairments, emphasizing its value for both clinical and exercise applications.

## 1. Introduction

In a medical, clinical, or practical context, new measurement methods should allow the collection of as much data of outstanding quality as possible [1]. When considering a new method as an alternative, it is crucial to systematically compare measurement techniques to ensure that technological or methodological advances do not compromise accuracy, reliability, or practical applicability, while identifying potential inconsistencies or limitations between methods [2,3]. The Halstead–Reitan Neuropsychological Test Battery is an extensive set of neuropsychological tests developed to evaluate diverse cognitive functions. This test battery is constructed to evaluate brain structure and functioning, aiming to identify potential impairments associated with neurological conditions, such as traumatic brain injuries, neurodegenerative diseases, and developmental disorders [4,5]. The tests were designed to monitor different cognitive domains, such as memory, attention, spatial processing, motor skills, and executive functions. The Halstead–Reitan Tests [6,7] offer a significant advantage due to their applicability across a wide age range, with specific versions for adults, children, and adolescents [8]. The ongoing relevance of inclusive measurement tools is vital in the field of healthcare, as they contribute to the development of effective interventions and improved patient care. Of those, the Finger Tapping Test is particularly effective in identifying fine motor control deficits [9], which is commonly used to assess neurodegenerative diseases such as Dementia, Alzheimer’s, and Parkinson’s [10,11], and has recently been applied in patients with fibromyalgia [12,13,14,15]. In previous studies, our research group has demonstrated that women with fibromyalgia exhibit distinct motor profiles compared to healthy controls, specifically showing increased variability in inter-tapping interval and decreased tapping frequency, reflecting deficits in motor synchronization and rhythm regulation [16,17]. These changes suggest an impaired ability to generate stable and coordinated motor patterns, possibly connected to alterations in the central sensory function and cortical excitability, which are typical in fibromyalgia patients. In this way, FTT emerges as a sensitive tool to capture subtle fine motor control deficits in this population. In this test, participants are instructed to tap a designated surface with the index finger as quickly as possible, for a fixed period of 10 s, for 5–10 trials per hand [4], assessing motor speed, coordination, and dexterity of the hands and fingers, both unilateral and bilateral [4,8]. Initially, the collected variables included the number of taps per trial (frequency), differences between dominant and non-dominant hands (laterality), and qualitative movement features, such as abnormal finger movements and disturbances of motor inhibition or apraxia [4,18,19,20]. Nonetheless, other variables can also be investigated, including range of motion or finger amplitude [21], fatigue effects [22], and inter-tap interval variability [23]. These variables can potentially reveal different movement patterns in clinical populations, proving to be particularly relevant in fibromyalgia, a central nervous system disorder characterized by fine and gross motor impairment [16]. Studies show that FM patients have limited movement, which can manifest as muscle coordination disorders and decreased hand function [13,24], highlighting the need to develop more sophisticated measurement methods that place greater emphasis on the movement process.

When substituting one measurement technique with another, it is imperative to employ statistical or analytical tools to evaluate inconsistencies [25], thereby confirming concordance between the new and established methods [1]. Correlation coefficients can be a useful measure to determine the strength of the relationship between two variables. However, there are three primary challenges in applying correlations to validation tests: First, correlation does not account for systematic differences (bias) between the two methods, where one method could consistently overestimate or underestimate values compared to the other, yet they could still be highly correlated [26]. For example, through one device, 20 more beats are counted systematically in all trials than through the other device. In this case, correlation could be close to 1.0, even though the two methods disagree in absolute terms. Second, correlation does not reflect the precision of the measurements. Even if two methods are highly correlated, one might have a much wider range of error or variability in its measurements than the other [27]. For example, the two systems may have similar mean values across all trials, but one device may show greater trial-to-trial variability. Finally, two time series could show a high correlation if they rank their measurements similarly, even when the actual magnitudes differ significantly [28] (i.e., if both devices detect the faster and slower participants in the same order, the correlation is higher, but there is a clear disagreement on the scale). The concern with the incorrect use of correlation coefficients led Bland and Altman [29] to develop the Bland–Altman technique, a graphical method that illustrates the agreement between two quantitative measurements by analyzing the mean difference and calculating the limits of agreement or confidence limits [1,30].

The inertial measurement units (IMUs), also known as inertial sensors, are small, portable biomechanical devices capable of collecting acceleration and angular velocity across three planes of motion with high frequency and accuracy [31]. Previous studies demonstrated their potential in evaluating motor control in the FTT [32,33]. In both studies, several inertial sensors were attached to the fingers and were validated with an optoelectronic system. However, the entire apparatus appears to restrict finger movements, which may distort accurate assessment, and also limits the concurrent qualitative analysis that healthcare practitioners have integrated into their standard practice. Furthermore, the use of the intraclass correlation coefficient may be insufficient to fully validate such systems. Nevertheless, the implementation of inertial sensors in the FTT can be simplified to enhance practicality, ease, and precision, in contrast to systems that require multiple sensors or 3D marker-based setups, or to methods/systems that only provide an outcome variable without capturing the execution process. Beyond preliminary studies, the use of wearables has been increasingly applied to assess fine motor skills, offering objective and high-resolution kinematic data. Maceira-Elvira et al. [34] highlighted the importance of wearables for clinical rehabilitation, particularly in the quantification of upper-limb motor function, outlining a roadmap for the integration of inertial sensors into clinical practice. Recently, Bremm et al. [35] showed that finger tapping performance based on accelerometry analysis provides a rigorous and automated quantification of motor deficits in neurological conditions. These findings reinforce the potential growth of inertial-based systems to complement observational methods and enhance the sensitivity of motor assessment in clinical and research contexts. The use of a single inertial sensor could enable the creation of an affordable system that is capable of collecting and computing multiple variables, such as the inter-tap interval, with a precision lower than 30 ms, well below the traditionally assumed reaction time [36]. It would also be possible to access and analyze the execution process by carrying out non-linear analyses, namely, multiscale entropy [16,37], to detect a loss of complexity patterns associated with various neurodegenerative or central nervous system disorders, aligning with the study’s focus on capturing the movement process, rather than solely the outcomes.

Given the ongoing demand for new measurement methods to gather higher-quality data [1], the growing relevance of portable devices such as IMUs, which offer substantial data collection capabilities, is evident [31]. Recent studies have demonstrated the use of inertial sensors to develop reliable solutions for tracking finger or hand movements [38,39], as well as for monitoring wrist motion during specific fine motor tasks in children [40,41]. Moreover, the use of inertial systems enables data processing techniques, such as data fusion and inertial signal transformation using convolutional kernels, to detect patterns that distinguish between experienced practitioners and beginners in specific martial arts motor skills [42].

The aim of this study is to determine the levels of agreement between two methods (videography and inertial sensors) for (i) the number of finger taps and (ii) the inter-tap interval in women with fibromyalgia, with the potential to enhance clinical monitoring and inform exercise strategies. We hypothesize that the validation of this new measurement method will lead to a deeper understanding of motor control in this population.

## 2. Materials and Methods

### 2.1. Sample

The data were collected from 20 women (aged 20–70 years), including 10 diagnosed with fibromyalgia and 10 healthy controls (Table 1). Each participant performed 12 trials (6 per hand), totaling 240 valid trials. The trials are repeated measurements nested by subject and hand.

Fibromyalgia participants were selected according to the following: (i) inclusion criteria: individuals diagnosed with fibromyalgia by a qualified healthcare professional following the American College of Rheumatology (ACR) guidelines; (ii) exclusion criteria: other diagnosed neurological and motor diseases. Control participants were selected according to the following: (i) inclusion criteria: individuals with physical characteristics identical to those of the fibromyalgia group, without diagnosed and associated diseases.

To ensure that both groups have comparable demographic characteristics, participants of the two groups were paired.

The project was approved by the Ethics Committee of the Santarém Polytechnic University (No. 2A-2022 ESDRM). All participants provided informed consent to participate in the study and were recruited through social platforms.

### 2.2. Procedures

Participants were asked to perform six trials of the FTT at their maximum speed for ten seconds per trial, starting with the preferred hand and repeating the entire process with the other hand. The implementation of 6 trials per hand was established based on the existing literature, which has typically employed 5 to 10 trials per hand, scoring the test by calculating the average number of taps on the surface from the best 5 trials [4,43]. Prior to the collection, participants were asked to demonstrate the task to ensure the instructions were understood.

To collect data for videography, a high-speed camera was used (Casio Exilim EX-ZR200, Casio Computer Co., Ltd., Tokyo, Japan), with a sample rate of 240 Hz. Simultaneously, an inertial sensor (MEMS type, TDK Invensense, model MPU9250, San Jose, CA, USA) was used to collect acceleration and angular velocity data at a sample rate of 100 Hz.

A rubber finger with a black dot was placed on the index finger to allow automatic digitization of the finger position in Kinovea software (version 2023.1.2) (Figure 1a) [44]. Volume calibration was used to calibrate the virtual space within the software (Figure 1b). Additionally, the inertial sensor was also placed on the index finger attached to the rubber finger (Figure 1c) to facilitate the simultaneous collection of three-dimensional linear acceleration and angular velocity. YAT software (version 2.4.1) [45] was employed to gather and record the data from the inertial sensor.

After digitizing the videos in the Kinovea software, the index position variable was extracted and imported to MATLAB R2021a [46], along with the raw data from the inertial sensor. Once the data was imported, a custom MATLAB routine was employed for analysis, enabling the synchronization of position data (obtained from Kinovea) with vertical linear acceleration data (acquired from IMU), as well as the identification of taps for both methods. Synchronization (Figure 2) was carried out by identifying the first tap on the surface in the finger’s vertical position graph (kinovea data) and the first acceleration peak in the finger’s vertical or linear acceleration graph (IMU data). For each trial, synchronization between videography (240 Hz) and the IMU (100 Hz) was performed using an event-based procedure: the first finger-to-surface contact was identified in each signal start as the reference point (*t* = 0). From that event, a 10 s window was extracted in each dataset according to the original timestamp, preserving the native sampling frequencies. The variables of interest (number of taps and inter-tap intervals) were computed from the detected peak times in each signal, without requiring point-by-point resampling. The temporal uncertainty associated with the sampling-rate difference is limited to half of the sampling interval (≈2 ms at 240 Hz; ≈5 ms at 100 Hz), which is negligible compared with the typical ITI duration (≈100–200 ms). The absence of relevant drift within the 10 s windows was confirmed by visual inspection of the overlaid traces.

Tapping automatic detection in MATLAB was applied using the *findpeaks* function to the time series of the vertical axis. Local maxima of the original signal and local minima of the inverted signal were identified. To reduce false positives, detection criteria were applied, including a minimum distance between peaks (0.120 s) and a minimum prominence of 20% of the total signal amplitude. These parameter values were defined based on preliminary tests with these IMU signals, as well as based on the expected physiological tapping frequency in adults (approximately 4–8 Hz, corresponding to inter-tap intervals of 0.125–0.250 s) [47,48]. Based on the recorded taps, the number of taps and the inter-tap intervals were calculated, along with their descriptive statistics (mean and standard deviation).

From the moment of the first tap, a time duration of 10 s was recorded for the task execution.

### 2.3. Data Treatment and Statistical Analysis

Data imported into the MATLAB environment was organized into a table, and the main variables (number of taps and inter-tap interval) were extracted. The first analysis involved calculating the mean and standard deviation across the six trials for each subject and hand (aggregation analysis). Next, an individual trial-level analysis was performed to assess intra-subject variation using repeated-measures and mixed-effects models.

Agreement between collection methods was evaluated using Bland–Altman analysis [25,29,49], with parametric or non-parametric approaches (bootstrap, 5000 repetitions), selected based on the distribution of differences [50].

The limits of agreement (LoAs), bias, and intraclass correlation coefficient (ICC (A, 1)) [51,52] were calculated with confidence intervals obtained by bootstrap [53,54]. Additionally, the adequacy of the sample was assessed using statistical power calculations for ICC and LoA precision and also by estimating the number of participants needed to achieve target widths for the agreement intervals [55,56]. All results and graphs were automatically exported in editable, high-resolution formats.

## 3. Results

The results of the various stages of data treatment are presented below. Figure 3 illustrates the finger positions after automatic digitization in Kinovea software and following import into a custom MATLAB routine.

The MATLAB routine successfully detected finger taps on the surface (red dots) and identified the maximum vertical position of the finger during each tap (green dots).

The finger acceleration plots from the IMU displayed less consistent patterns for identifying surface taps compared to the position plots. For example, in Figure 4, the peak indicated by the circle usually corresponds to the second peak of the tap cycle; however, in this instance, the first peak of the cycle was identified instead.

After data preparation, the agreement between Kinovea and IMU data was assessed, for both the number of taps (Figure 5) and the inter-tap interval (Figure 6). The agreement analysis was conducted using data aggregated by Subject×Hand, calculated as the average of six trials per hand for each subject. This approach reduces intra-subject variability and provides a summary value for each condition. Table 2 presents the main results of the agreement analysis between IMU and videography.

For the number of taps, the normality was not assumed (Lilliefors *p* = 0.036), leading to a non-parametric Bland–Altman analysis, which has shown a median bias of −0.33 taps with a limit of agreement (LoA) between −6.27 and 6.33 taps (Figure 5). The reproducibility coefficient (RPC_np_) was 4.3 taps, and the intraclass correlation coefficient (ICC_A,1_) was 0.94 (95% CI: 0.89–0.96), assumed as an excellent agreement.

Normality was not assumed for the inter-tap intervals (Lilliefors *p* < 0.001). The median bias was 0.005 s, with the non-parametric LoA between −0.042 and 0.061 s (Figure 6). The reproducibility coefficient was 0.02 s, and the agreement was considered excellent with an ICC = 0.89 (95% CI: 0.83–0.94).

The regression analysis included in Bland–Altman did not show any significant proportional trends for all variables (number of taps: slope = 0.013, *p* = 0.825; ITI: slope = 0.079, *p* = 0.285).

The agreement limits were calculated using both the classic method (average ± 1.96 standard deviation), as recommended by Bland and Altman (1986) [27], and a non-parametric bootstrap method (5000 repetitions), which is appropriate when the differences do not follow a normal distribution. Presenting both methods allows comparison of the traditional results with robust estimations, reinforcing the validity of the analysis. To complement this, the differences between methods were evaluated by comparing parametric and non-parametric limits of agreement (LoAs) (bootstrap, 5000 repetitions). For the number of taps, the parametric limits ranged from −6.09 to 4.44 taps, while the non-parametric limits ranged from −6.27 to 6.33 taps (Figure 7).

For ITI, the limits of agreement for the parametric method oscillated between −0.031 and 0.046 s, and for the non-parametric method, they oscillated between −0.042 and 0.061 s (Figure 8). In both cases, the distributions show a bias close to zero and narrow confidence intervals, supporting the robustness of the results.

When all the 238 valid individual trials (trial-level analysis) are analyzed through mixed-effects models, the results show that IMU tends to underestimate the values. Considering the number of taps, the mean bias was −12.93 taps, with a significant bias (slope = 0.263, *p* = 1.18 × 10^−6^) and a residual intra-subject variability of 2.07. For ITI, the bias was −0.055 s, also with a significant proportional bias (slope = 0.283, *p* = 3.46 × 10^−8^) and a residual variability of 0.013 s.

Additionally, sample size adequacy was also tested for the ICC analysis. Power calculations indicated that a minimum of 11 participants would be required to detect an ICC of 0.90 against a minimum acceptable ICC of 0.75, with α = 0.05 and 80% power. Since the study included 20 participants, the sample size was sufficient to ensure adequate statistical power for the agreement analysis based on ICC.

## 4. Discussion

This validation study assessed the levels of agreement between videography (Kinovea) and an inertial sensor (IMU). Similar levels of agreement have been reported in studies using IMU-based assessments of fine motor performance. In the study by Bremm, Pavelka, Garcia, Mombaerts, Krüger and Hertel [35], the ICC values ranged from 0.85 to 0.95 for tapping-related sub-items of the MDS-UPDRS. Our ICC values between 0.89 and 0.94 are consistent with that study, showing excellent agreement for the number of taps and good-to-excellent agreement for inter-tap interval, specifically for Subject×Hand aggregation, with a median bias close to zero and a high intra-correlation coefficient. However, in a trial-level analysis, a proportional trend indicating systemic underestimations by the IMU was verified, mostly for tap counting. This should be considered for interpretation and for future algorithm design [29,52,57].

Based on the results of this study, most of the taps detected on the surface by the new method using acceleration data from IMU fell within the confidence interval, demonstrating strong consistency across methods. The inclusion of both parametric and bootstrap-derived LoA strengthens the evidence for agreement at the aggregated level. In this type of analysis, the bias in the number of taps and the inter-tap interval was close to zero, and the limits of agreement aligned with the 95% confidence interval. Additionally, the reproducibility coefficient (RPCnp) indicated very small values, especially for ITI, suggesting that the traditional and the new methods exhibit great similarity, allowing good agreement between them [58]. These findings are consistent with those reported by Morrow et al. [59], who also found minimal discrepancies between IMU-based measurements and standard clinical assessments for upper-limb motor tasks, highlighting the IMU’s potential for accurate movement quantification. Furthermore, the claim that a single inertial sensor can effectively capture the number of finger taps during the FTT contributes to the growing body of evidence suggesting that such technology can enhance the precision of motor disorder evaluations [16,60] and holds significant potential for developing assessment and analysis systems for special populations.

In this study, we employed a validation approach using 2D videography analysis, which restricts motion assessment to a single plane, in order to minimize resource requirements. However, the use of three-dimensional analysis—whether through videography or other more sophisticated systems—would entail considerable time and financial resources for a test that is traditionally low-cost and easy to implement.

Unlike traditional counting methods, the IMU-based approach provides a rich dataset from which nuanced analyses of motor function can be conducted, offering insights into the subtle motor deficits that characterize various neurological conditions [61]. For instance, the IMU accurately captured inter-tap intervals with high ICC and low RPCnp, which are consistent with the results of Kim et al. [62], who demonstrated that wearable IMUs can reliably quantify fine motor control in stroke and in Parkinson’s disease.

The inclusion of patients with fibromyalgia enabled the evaluation of these instruments within a population with specific needs, regarding the attachment of devices to body segments, and characterized by a reluctance to engage in extensive movement and often exhibiting high levels of fatigue. Furthermore, it provided the opportunity to collect data from a cohort displaying central nervous system characteristics that have implications for motor function. Although comparing the groups incorporated into this study was not an objective, the excellent agreement demonstrates that IMUs can be used effectively even in populations with impaired motor control. We recognize that this approach facilitated a comprehensive assessment of performance variability on the Finger Tapping Test, highlighting the versatility and adaptability of IMU technology, which can be tailored to meet the specific needs of different patient groups, thereby enhancing the inclusivity and accessibility of motor function evaluations in clinical practice. Bisi and Stagni [40] and Liu, Qiu, Wang, Dong, and Yu [41] assessed wrist motion in children without autism and children with autism, respectively, demonstrating once more that IMU-based systems can quantify fine motor performance and discriminate between development profiles. These findings reinforce the potential of inertial sensors to capture subtle motor control differences across populations with distinct neurological or developmental conditions, supporting their use in both research and clinical assessment contexts.

This study, allied to the feasibility of integrating IMUs into wearable technologies, further enhances their utility, promising continuous and unobtrusive monitoring of motor functions. These advancements could contribute to the earlier identification and monitoring of neurodegenerative conditions by providing detailed information on motor patterns over time. Furthermore, by employing sophisticated analytical tools such as machine learning algorithms, it may be possible to extract meaningful insights from these data, enabling the characterization of motor deficits and the identification of biomarkers for various neurological conditions.

Based on our results, IMU-based methods show promising applications in the field of neurorehabilitation, particularly by enabling continuous monitoring of motor functions outside of clinical settings. The ability that IMU devices have to accurately quantify movements suggests that they could support the design of personalized and adaptive exercises, improving the precision of real-time feedback and enhancing motor learning outcomes. Additionally, the integration of IMUs into telemedicine platforms may enhance accessibility for patients with limited mobility or those living in remote areas.

While the use of IMU technology for motor performance assessment in healthcare has undeniable benefits, it is also crucial to recognize the challenges and limitations associated with it. Variability in sensor placement, potential signal noise, and proper data interpretation all require thorough consideration to guarantee the precision and dependability of measurements.

In this study, data processing involves two types of analysis. The aggregate analysis showed that using a traditional approach, tap detection and inter-tap interval have great application power. However, when a trial-level repeated-measures mixed-models approach was applied—which offers a more realistic and clinically relevant perspective by accounting for each individual repetition and revealing potential limitations not apparent in the aggregate analysis—a systematic and proportional bias was found for both variables under study. From a practical point of view, this suggests that the implementation of IMUs to capture the number of taps or the inter-tap interval may be used, especially at the aggregate level of analysis. Regarding the use of trial-level implementation, it should focus, in the future, on refining signal processing procedures—namely, filtering parameters and adaptive peak detection thresholds—and possibly include calibration trials or machine learning algorithms to automatically adjust sensitivity based on the individual’s tapping time. Such adjustments could reduce proportional bias and enhance the usability of IMUs for detailed motor performance assessment. Such adaptive models have shown promising results in upper-limb motor assessment, as demonstrated by Bremm, Pavelka, Garcia, Mombaerts, Krüger, and Hertel [35], who achieved >90% accuracy in movement-feature detection using IMU-based machine learning. Our results indicate that the application of these instruments requires further development, particularly in defining automatic tap detection parameters and optimizing filter characteristics, both of which are common considerations in signal processing.

Beyond fibromyalgia, the methodological framework applied in this study, combining IMU-based kinematic data with automated signal processing and agreement analysis, can be extended to other populations characterized by altered motor control. IMU-based methods have been successfully used in Parkinson’s disease, multiple sclerosis, post-stroke rehabilitation, and developmental coordination disorder [62,63]. Such conditions share common challenges in fine motor assessment and coordination, suggesting that IMU-based assessments may provide sensitive and quantitative markers of dysfunction applicable across diverse clinical and rehabilitation settings [35,64,65,66].

Additionally, the adoption of these technologies raises ethical and privacy concerns regarding continuous individual monitoring. Such concerns should, however, be allayed with strict privacy measures and ethical guidelines to ensure the responsible use of inertial sensors in healthcare.

Finally, the robustness of these results was also tested with a posterior power analysis; although the sample was modest, this analysis indicated that 11 participants would have been sufficient to detect excellent agreement (ICC ≥ 0.90 vs. minimum acceptable ICC = 0.75) with 80% power. Therefore, the inclusion of 20 participants ensured sufficient power for the ICC analysis, even if a larger sample would be desirable to further reduce the uncertainty of the limits of agreement for tap counts.

## 5. Conclusions

This research delved into the potential of incorporating inertial measurement units (IMUs) as a novel way to evaluate motor function through the Finger Tapping Test (FTT). Through comparing traditional evaluation methods and traditional statistical analysis with IMU-based measurements, a strong level of congruence was established with minimal discrepancies observed. Additionally, the IMU-based approach not only precisely captured the amount of finger taps but also provided a wealth of data for in-depth analysis of motor function, potentially refining our understanding of motor disorders. By including patients with fibromyalgia, the efficacy of IMU technology was further showcased within a specific group with distinct motor function needs, showcasing its versatility and adaptability.

Moreover, the ability to continuously monitor motor functionalities beyond traditional clinical environments presents exciting possibilities in neurorehabilitation and telemedicine. By implementing the protocol of FTT, appropriate precautions, and ethical standards, the use of IMU technology presents a groundbreaking opportunity to transform the evaluation and treatment of motor functions into clinical settings. However, challenges still exist regarding the consistency of sensor placement, automatic detection parameters, interference from signal noise, the systematic and proportional bias found in trial-level analysis, and the ethical implications of continuous monitoring. To face these challenges, future development should focus on refining automatic signal processing algorithms to advance tap detection accuracy. Adaptive filtering, machine learning-based threshold adjustment, and calibration routines tailored to individual motor profiles may enhance the robustness and generalizability of IMU-based assessments, which have been applied not only in FTT but also to other tasks, namely, walking gait [35,67].

This study established a concurrent or criterial validity in IMU-based measures compared to videography during the Finger Tapping Test, with a high agreement level both for the number of taps and for the inter-tap interval. The IMU’s ability to capture the core constructs of the task in a reliable way further supports its construct validity; even if the predictive and ecological validity were not assessed, the inclusion of a clinical group (fibromyalgia) shows the possibility to apply these results and reinforces the potential use of IMUs in different populations.

As we continue to delve into the possibilities of IMU technology, there is potential for further investigation into its integration with advanced analytical tools like machine learning algorithms. Based on this validation, future work will focus on the integration of IMU data with machine learning algorithms to automatically identify motor patterns and predict functional impairments. This represents a concrete step toward developing robust, data-driven assessment tools for clinical and exercise contexts, enabling objective monitoring and early detection of motor dysfunction. Through this, we are able to unlock new paths for identifying biomarkers linked to different neurological conditions and customize rehabilitation techniques. In essence, IMU-based approaches present an exciting development in evaluating motor function, presenting opportunities for improved diagnostics, tailored interventions, and, ultimately, better outcomes for patients.

## Figures and Tables

**Figure 1 sports-13-00373-f001:**
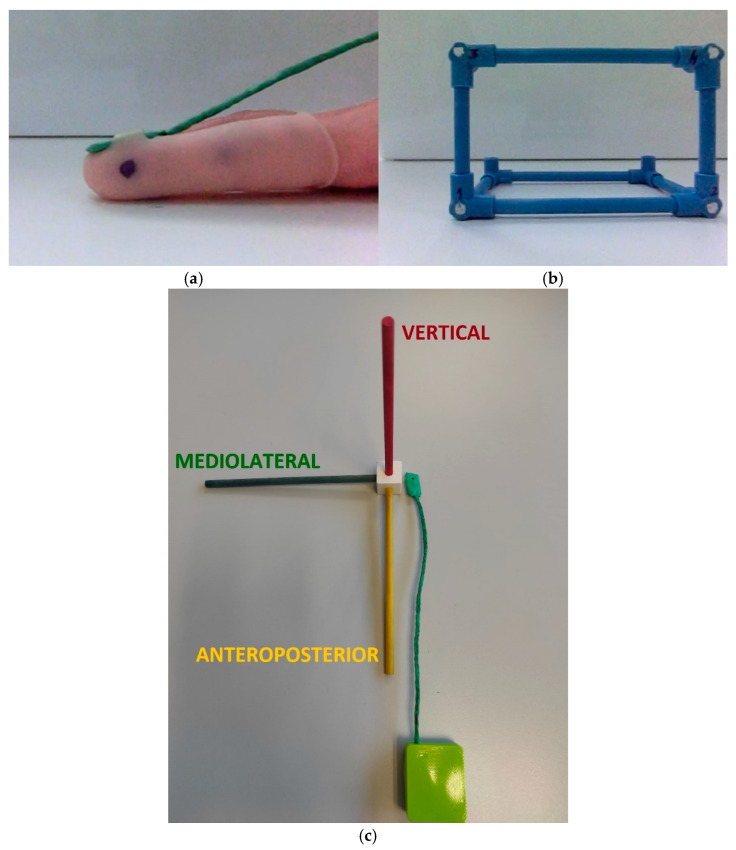
(**a**) FTT index finger; (**b**) Kinovea calibration volume; and (**c**) inertial sensor 3D axis.

**Figure 2 sports-13-00373-f002:**
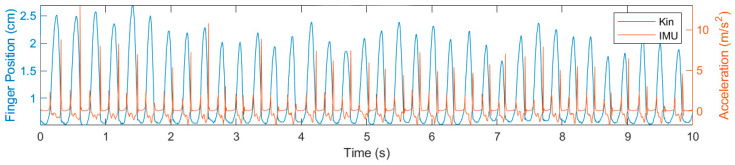
Synchronization between the time series of Kinovea (Kin) and IMU data.

**Figure 3 sports-13-00373-f003:**
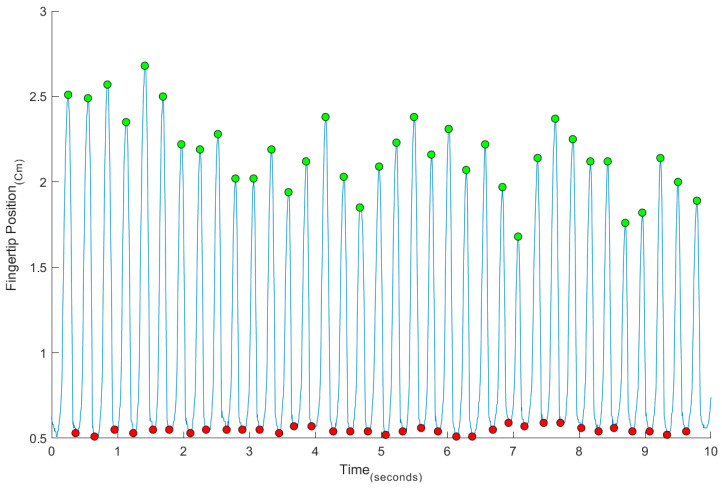
Example of a temporal series of FTT finger position (red dots—taps on the surface; green dots—maximum vertical position).

**Figure 4 sports-13-00373-f004:**
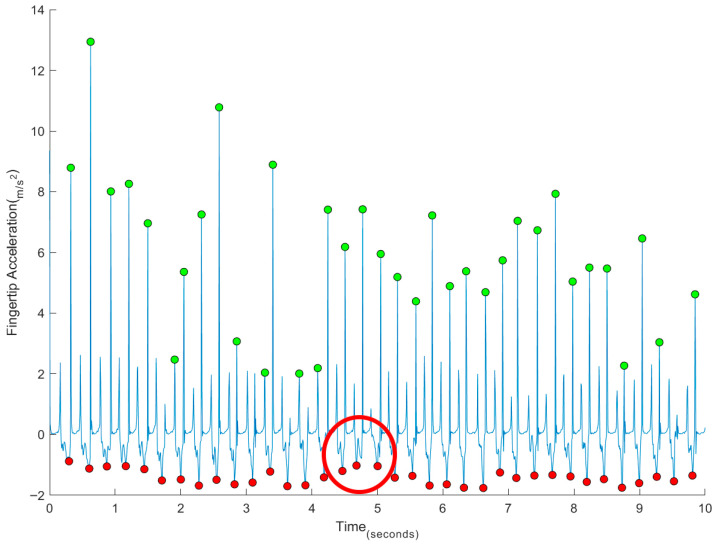
Example of a time series of FTT linear acceleration, with cases of detection limitations (red circle).

**Figure 5 sports-13-00373-f005:**
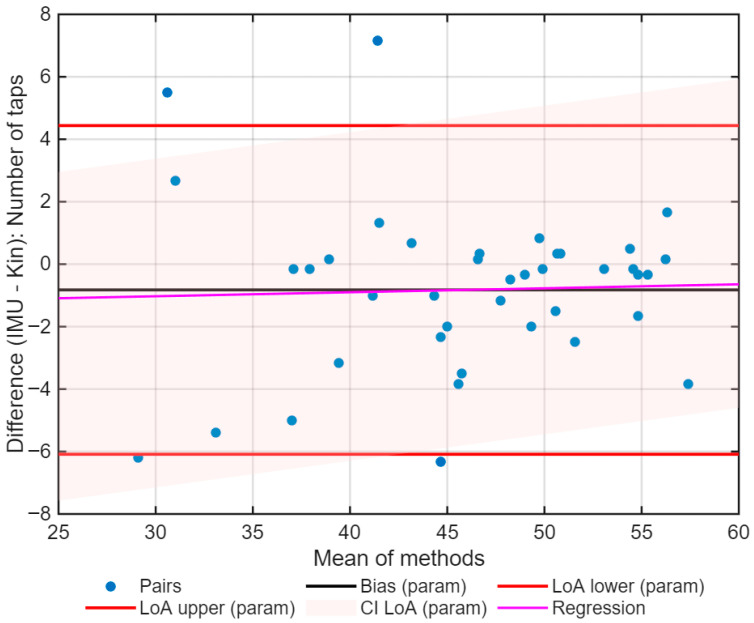
Bland–Altman plot for the number of taps during the Finger Tapping Test (FTT), comparing Kinovea and IMU methods. The blue dots are the individual differences for the average of the two methods. The black line shows the bias, the red lines show the limits of agreement, and the pink shadow area shows the respective confidence interval. The magenta line corresponds to linear regression.

**Figure 6 sports-13-00373-f006:**
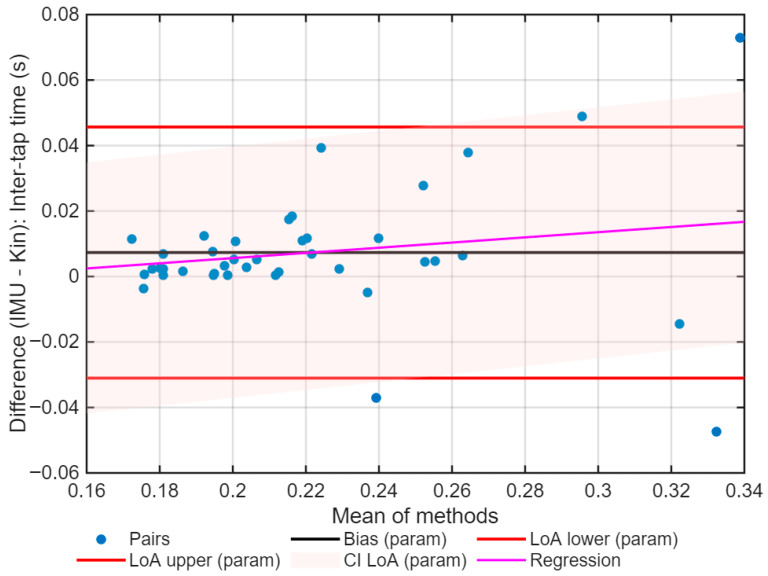
Bland–Altman plot for the inter-tap interval during the Finger Tapping Test (FTT), comparing Kinovea and IMU methods. The blue dots are the individual differences for the average of the two methods. The black line shows the bias, the red lines show the limits of agreement, and the pink shadow area shows the respective confidence interval. The magenta line corresponds to the linear regression.

**Figure 7 sports-13-00373-f007:**
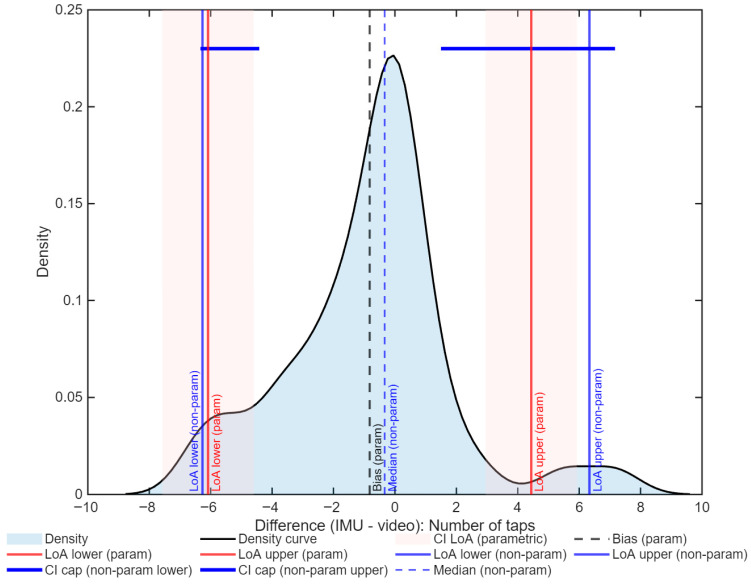
Distribution of differences between IMU and video (Kinovea) measurements for the number of taps in the Finger Tapping Test. The black line represents the density curve, and the blue area corresponds to the distribution of differences. The red lines indicate the parametric limits of agreement (LoAs) with their respective confidence intervals (shaded in red). The blue lines represent the non-parametric limits of agreement (bootstrap, 5000 repetitions), with the horizontal bars marking the 95% confidence intervals. The dashed lines indicate the mean bias (parametric, black) and the median (non-parametric, blue).

**Figure 8 sports-13-00373-f008:**
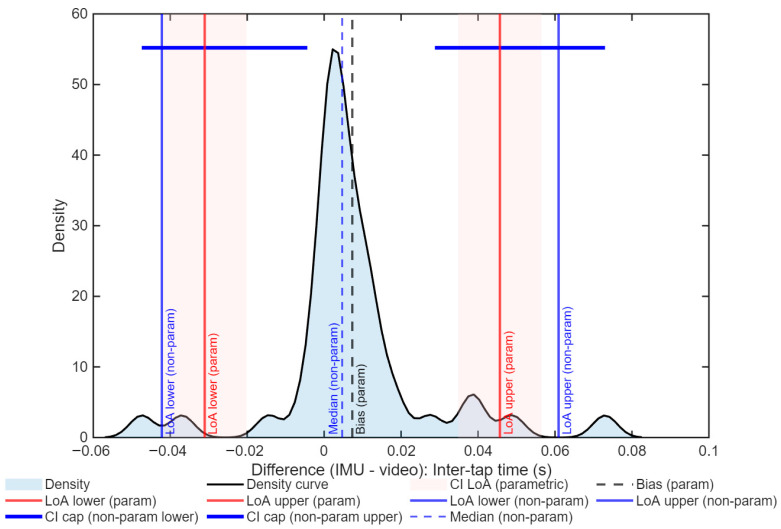
Distribution of differences between IMU and video (Kinovea) for the inter-tap interval (ITI) in the Finger Tapping Test. The black line represents the density curve, and the blue area corresponds to the distribution of differences. The red lines indicate the parametric limits of agreement (LoAs) with their respective confidence intervals (shaded in red). The blue lines represent the non-parametric limits of agreement (bootstrap, 5000 repetitions), with the horizontal bars marking the 95% confidence intervals. The dashed lines indicate the mean bias (parametric, black) and the median (non-parametric, blue).

**Table 1 sports-13-00373-t001:** Sample characterization.

Group	Age		Height		Weight	
	Mean	SD	Mean	SD	Mean	SD
Fibromyalgia	46.400	12.714	162.900	5.243	63.000	10.536
Control	45.900	12.950	157.800	5.671	60.700	5.675
Total	46.150	12.835	160.350	6.027	61.850	8.540

SD—standard deviation.

**Table 2 sports-13-00373-t002:** Results of the agreement analyses between IMU and videography (Kinovea) in the Finger Tapping Test (FTT).

Variable	Number of Taps	Inter-Tap Time (s)
Normality (Lilliefors *p*)	0.036 (rejected)	<0.001 (rejected)
Selected method	Non-parametric	Non-parametric
Bias	−0.33	0.005
LoA (95%)	−6.27 to 6.33	−0.042 to 0.061
ICC(A,1) [95% CI]	0.94 [0.89–0.96]	0.89 [0.83–0.94]
Mixed-effects bias	−12.93	−0.055
Slope (*p*)	0.263 (*p* < 0.001)	0.283 (*p* < 0.001)
σ resid	2.07	0.013

Note: The aggregate analysis (Subject × Hand) included tests for normality (Lilliefors), selected method (parametric vs. non-parametric), mean bias (Bias), limits of agreement (LoA, 95%), and intraclass correlation coefficient [ICC(A,1), with 95% CI]. The trial-level analysis, using mixed-effects models, shows the average bias, regression slope (slope, with respective *p*-value), and intra-subject residual variability (σ resid).

## Data Availability

The data are not publicly available due to ethical restrictions. However, portions of the dataset may be made available upon reasonable request and subject to approval by the institutional ethics committee. The ethics committee only authorized data sharing among the researchers responsible for the study.

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
