# Peer review of "Precision of an Inertial System to Evaluate the Finger Tapping Test in Women with Fibromyalgia"

_sports, 2025, doi:10.3390/sports13110373_

Round 1

Reviewer 1 Report

Comments and Suggestions for Authors

Dear Authors,

Your study presents an important and well-conceived contribution to the validation of IMU-based assessment methods in clinical and exercise contexts. The manuscript is interesting and coherent, but some sections need to be clarified and reinforced. Below are my specific observations.

General Comments

Abstract:

1) The phrase “with only a very small difference…” should be reformulated for clarity and tone. Suggested alternatives: “showing minimal differences” or “with negligible bias.”

2) The expression “clinical and exercise contexts” is too generic. Please specify which contexts are meant (e.g., rehabilitation, physiotherapy, home-based monitoring).

Keywords:

1) Use full terms instead of acronyms where possible.

Introduction:

1) Clarify how the Finger Tapping Test differentiates motor skills in women with fibromyalgia. Which variables are most sensitive, and what is the clinical rationale?

2) Expand the literature review and strengthen references. Several studies have already combined the tapping test with wearable sensors to provide more objective data, which would reinforce the introduction.

Bibliography:

1) Some references lack DOIs. Please add them when available.

Line-by-Line Comments

Lines 36–37: The sentence beginning with “ensuring that advancements…” is awkward. The second part (“pinpointing inconsistencies...”) is disconnected. Please revise for clarity.

Line 38: Add a reference when introducing “The Halstead-Reitan Neuropsychological Test Batteries.”

Lines 70–72: The three critiques of correlation use are valid but abstract. Provide short examples to clarify each limitation.

Line 84: Remove the period after “accuracy.”

Line 92: The statement that IMU use “can be simplified” is interesting but vague. Specify how the simplification was implemented.

Lines 119–122: Indicate whether participants were given a familiarization period before testing.

Lines 124–127: The sampling rate difference (240 Hz for the camera and 100 Hz for the IMU) could affect synchronization. Clarify whether this was addressed.

Lines 140–142: Clarify whether synchronization between Kinovea and IMU was manual or automatic and indicate the margin of error.

Lines 144–146: Justify the parameters used for automatic tap detection (e.g., minimum peak distance of 0.120 s). Were they based on literature, preliminary testing, or chosen arbitrarily?

Lines 230–231: The non-parametric limits of agreement for tap count appear inconsistent (“6.27 to 6.33”). Probably it should be “–6.27 to 6.33.” Please check.

Lines 321–325: The claim that IMUs could enable early diagnosis through wearable technology should be expressed more cautiously or supported with references.

Lines 344–348: The systematic bias found at the trial level is mentioned but not discussed in depth. Explain how this affects IMU usability and what improvements could reduce it.

Lines 378–380: Strengthen the conclusion by emphasizing future development of more robust automatic algorithms. Support this point with relevant recent references.

Lines 388–389: Discuss the potential generalizability of the method to other clinical populations with motor impairments.

Lines 390–392: The reference to machine learning is too superficial. Clarify whether this is a concrete future direction or only a theoretical possibility.

Author Response

Dear reviewer, thanks for your suggestions, in fact it makes the manuscript easier to read and understand.

Below, we leave the responses to each comment.

Reviewer 1

Comments and Suggestions for Authors

Dear Authors,

Your study presents an important and well-conceived contribution to the validation of IMU-based assessment methods in clinical and exercise contexts. The manuscript is interesting and coherent, but some sections need to be clarified and reinforced. Below are my specific observations.

General Comments

Abstract:

1) The phrase “with only a very small difference…” should be reformulated for clarity and tone. Suggested alternatives: “showing minimal differences” or “with negligible bias.”

Response: The suggestion was followed. Since it concerns the difference between beats, the text in question was replaced by “showing minimal differences”.

2) The expression “clinical and exercise contexts” is too generic. Please specify which contexts are meant (e.g., rehabilitation, physiotherapy, home-based monitoring).

Response: It was added more specific information.

Keywords:

1) Use full terms instead of acronyms where possible.

Response: The acronyms IMU and FTT were replaced by the full terms.

Introduction:

1) Clarify how the Finger Tapping Test differentiates motor skills in women with fibromyalgia. Which variables are most sensitive, and what is the clinical rationale?

Response: In the introduction section was added information regarding previous studies with fibromyalgia patients (ln. 55-62).

2) Expand the literature review and strengthen references. Several studies have already combined the tapping test with wearable sensors to provide more objective data, which would reinforce the introduction.

Response: Added information regarding this comment was added in lines 102-111 of the introduction section.

Bibliography:

1) Some references lack DOIs. Please add them when available.

Line-by-Line Comments

Lines 36–37: The sentence beginning with “ensuring that advancements…” is awkward. The second part (“pinpointing inconsistencies...”) is disconnected. Please revise for clarity.

Response: The sentence has been rewritten to clarify the content.

Line 38: Add a reference when introducing “The Halstead-Reitan Neuropsychological Test Batteries.”

Response: Although there was a bibliographical reference to this battery of tests at the end of the sentence, references from the original manual and a primary source from the adult version were added. 

Lines 70–72: The three critiques of correlation use are valid but abstract. Provide short examples to clarify each limitation.

Response: Brief examples have been added for each limitation in the use of correlation.

Line 84: Remove the period after “accuracy.”

Response: Removed.

Line 92: The statement that IMU use “can be simplified” is interesting but vague. Specify how the simplification was implemented.

Response: Information has been added to address the reviewer's suggestion.

Lines 119–122: Indicate whether participants were given a familiarization period before testing.

Response: A sentence was added to clarify the familiarization and understanding of the task .

Lines 124–127: The sampling rate difference (240 Hz for the camera and 100 Hz for the IMU) could affect synchronization. Clarify whether this was addressed.

Response: A paragraph was added to clarify the method used.

Lines 140–142: Clarify whether synchronization between Kinovea and IMU was manual or automatic and indicate the margin of error.

Response: This question was included in the previous added paragraph.

Lines 144–146: Justify the parameters used for automatic tap detection (e.g., minimum peak distance of 0.120 s). Were they based on literature, preliminary testing, or chosen arbitrarily?

Response: An explanation of the parameters used to automatic tap detection were added in procedures section. It included preliminary testing and also, they are based on literature.

Lines 230–231: The non-parametric limits of agreement for tap count appear inconsistent (“6.27 to 6.33”). Probably it should be “–6.27 to 6.33.” Please check.

Response: Thank you for pointing that out. It really is -6.27. Corrected.

Lines 321–325: The claim that IMUs could enable early diagnosis through wearable technology should be expressed more cautiously or supported with references.

Response: The text has been rewritten to clarify that this is a future possibility for the use of this type of device.

Lines 344–348: The systematic bias found at the trial level is mentioned but not discussed in depth. Explain how this affects IMU usability and what improvements could reduce it.

Response: Text has been added to indicate the implications of the results that improvements may be made in the future to reduce this bias.

Lines 378–380: Strengthen the conclusion by emphasizing future development of more robust automatic algorithms. Support this point with relevant recent references.

Response: Text has been added to comply with this comment.

Lines 388–389: Discuss the potential generalizability of the method to other clinical populations with motor impairments.

Response: In the discussion section, a paragraph has been added to address this issue.

Lines 390–392: The reference to machine learning is too superficial. Clarify whether this is a concrete future direction or only a theoretical possibility.

Response: Added information, of a future direction of our research group.

Reviewer 2 Report

Comments and Suggestions for Authors

Interesting idea of ​​this study, my recommendations are the following:
Abstract – I recommend mentioning the age of the subjects. Line 22 I recommend mentioning mathematical values ​​that support the statement.
Lines 32-37 I recommend rewriting as a sentence. I recommend expanding the section.
I recommend presenting the motor aspects specific to fibromyalgia in more detail in the Introduction.
Line 95 I recommend deleting – primarily, because you do not present the secondary aims.
2. Materials and Methods – I recommend introducing a new subsection called Study design where the typology of the study and other specific aspects should be mentioned.
Lines 107-108 are repeated in lines 103-104 I recommend deleting.
Lines 104-106 and 108-110 I recommend moving to the Procedures section and rewriting for clarification.
Section 2.1. I recommend mentioning the inclusion or exclusion criteria.
Section 2.3. I recommend mentioning that the anthropometric data concerned X and SD.
Lines 342-351 are findings and sound like a plea, I recommend revision.
Lines 259-263 recommend moving to section 2.1.
Lines 532-355 recommend deleting, it is not relevant in this section, possibly to be mentioned in the Limitations of the study. I recommend mentioning at the end of the Discussion section what are the limits of this study as well as future research directions.
Lines 305-329 recommend moving to the Introduction or Conclusion section without duplicating information. The aspects mentioned are presented as a plea regarding the purpose of this study, I recommend major revision.
Lines 356-361 recommend deleting, the information is duplicated.
In conclusion, I recommend that the Discussion section be revised significantly, by making concrete comparisons between the results of this study and those of previous studies, based on the specialized literature.
I recommend revising the conclusions, focused on the results, they are too long and general.
Bibliographic indices 32,33,34 I recommend mentioning them in more detail and adding links.

Author Response

Dear reviewer, thanks for your suggestions, in fact it makes the manuscript easier to read and understand.

Below, we leave the responses to each comment.

Reviewer 2

Interesting idea of ​​this study, my recommendations are the following:

Abstract – I recommend mentioning the age of the subjects.

Response: This information was added.

 Line 22 I recommend mentioning mathematical values ​​that support the statement.

Response: This information was added.

Lines 32-37 I recommend rewriting as a sentence. I recommend expanding the section.
I recommend presenting the motor aspects specific to fibromyalgia in more detail in the Introduction.

Response: This information was added.

Line 95 I recommend deleting – primarily, because you do not present the secondary aims.

Response: Removed.

  1. Materials and Methods – I recommend introducing a new subsection called Study design where the typology of the study and other specific aspects should be mentioned.

Response: Additional information about the study design has been included in the Materials and Methods section, according to the suggestions of both reviewers.  

Lines 107-108 are repeated in lines 103-104 I recommend deleting.

Response: Removed.

Lines 104-106 and 108-110 I recommend moving to the Procedures section and rewriting for clarification.

Response: Thank you for your suggestion but we decided to keep this information in the Sample section because, for this type of analysis, the 240 trials represent the sample of trials included in the trial-level analysis, therefore, it provides essential context about the sample size.

Section 2.1. I recommend mentioning the inclusion or exclusion criteria.

Response: This information was added.

Section 2.3. I recommend mentioning that the anthropometric data concerned X and SD.

Response: Anthropometric data were not reported because the analysis specifically focused on the variables related to the taps, which were assessed using Bland-Altman plots and ICC.

Lines 342-351 are findings and sound like a plea, I recommend revision.

Response: The text has been adapted to meet the reviewer's suggestion.

Lines 259-263 recommend moving to section 2.1.

Response: We understand the recommendation of the reviewer, probably based on the initial sentence of the paragraph. We decided to keep the text in this location because the opening sentence serves as an introduction to the results presented regarding trial-level mixed models, therefore should be in results section and not in the methods-sample sample section.

Lines 532-355 recommend deleting, it is not relevant in this section, possibly to be mentioned in the Limitations of the study. I recommend mentioning at the end of the Discussion section what are the limits of this study as well as future research directions.

Response: We apologize, but we do not understand which lines the reviewer is referring to. Lines 532-355 do not exist in the manuscript.

Lines 305-329 recommend moving to the Introduction or Conclusion section without duplicating information. The aspects mentioned are presented as a plea regarding the purpose of this study, I recommend major revision.

Response: The discussion was revised and some sentences were moved to introduction and in some cases were deleted when the information was redundant.

Lines 356-361 recommend deleting, the information is duplicated.

Response: This recommendation associated with a comment of the other reviewer led to a better explanation and reflection of the information stated.

In conclusion, I recommend that the Discussion section be revised significantly, by making concrete comparisons between the results of this study and those of previous studies, based on specialized literature.

Response: Thank you very much for this suggestion. The authors reviewed the discussion and rewrote it to more clearly indicate the results of specialised studies in the literature. We believe it is now clearer and specific.

I recommend revising the conclusions, focused on the results, they are too long and general.

Response: Regarding this comment, we tried to reduce the text, but focusing on four parts, namely the summary of results, practical interpretation, future projection/development, and general discussion on applicability. We also took into account the comments from the other reviewer.

Bibliographic indices 32,33,34 I recommend mentioning them in more detail and adding links.

Response: References 32, 33 and 34 corresponds to Kinovea, YAT, and MATLAB, respectively. As these are software applications used for data collection, processing and analysis, their references are necessarily limited in detail. They are cited specifically to acknowledge the tools employed in the study.

Round 2

Reviewer 1 Report

Comments and Suggestions for Authors

Dear Authors,

Thank you for your comprehensive revision. You have effectively addressed most of the comments, resulting in a manuscript that is clearer, more coherent, and scientifically robust. I particularly appreciated the additional methodological details and the enhanced discussion, both of which significantly contribute to the overall quality of the paper. Before we reach final acceptance, I recommend two additional improvements. First, please ensure that all references include their DOI, where available, as several are still missing. Second, it would be beneficial to incorporate more recent and specific studies on the application of IMUs in several contexts, rather than relying on general references to wearable devices. Strengthening the introduction and discussion with literature that closely aligns with your proposed approach would add further depth and credibility to your argument.

Overall, this is a valuable and well-crafted contribution. I look forward to reviewing the final version.

Best regards

Author Response

Dear reviewer, thanks for your suggestions, in fact it makes the manuscript easier to read and understand.

Below, we leave the responses to each comment.

Reviewer 1

Comments and Suggestions for Authors

First, please ensure that all references include their DOI, where available, as several are still missing.

Response: We appreciate the reviewer’s comment. The missing DOIs have been added to the references whenever available.

Second, it would be beneficial to incorporate more recent and specific studies on the application of IMUs in several contexts, rather than relying on general references to wearable devices. Strengthening the introduction and discussion with literature that closely aligns with your proposed approach would add further depth and credibility to your argument.

Response: Thank you very much for your comment. It were added new references showing the different applications of systems based on IMUs, both in introduction and in discussion.

Reviewer 2 Report

Comments and Suggestions for Authors

No comments

Author Response

We thank the reviewer for the valuable suggestions, which greatly improved our manuscript.